# Claudin-2-dependent paracellular channels are dynamically gated

Christopher R Weber[1], Guo Hua Liang[1], Yitang Wang[1], Sudipto Das[1†], Le Shen[1], Alan S L Yu[2,3], Deborah J Nelson[4], Jerrold R Turner[1,5]*

[1]Department of Pathology, The University of Chicago, Chicago, United States; [2]Division of Nephrology and Hypertension, University of Kansas Medical Center, Kansas City, United States; [3]Kidney Institute, University of Kansas Medical Center, Kansas City, United States; [4]Department of Pharmacological and Physiological Sciences, The University of Chicago, Chicago, United States; [5]Departments of Pathology and Medicine (GI), Brigham and Women's Hospital and Harvard Medical School, Boston, United States

**Abstract** Intercellular tight junctions form selectively permeable barriers that seal the paracellular space. Trans-tight junction flux has been measured across large epithelial surfaces, but conductance across individual channels has never been measured. We report a novel trans-tight junction patch clamp technique that detects flux across individual claudin-2 channels within the tight junction of cultured canine renal tubule or human intestinal epithelial monolayers. In both cells, claudin-2 channels display conductances of ~90 pS. The channels are gated, strictly dependent on claudin-2 expression, and display size- and charge-selectivity typical of claudin-2. Kinetic analyses indicate one open and two distinct closed states. Conductance is symmetrical and reversible, characteristic of a passive, paracellular process, and blocked by reduced temperature or site-directed mutagenesis and chemical derivatization of the claudin-2 pore. We conclude that claudin-2 forms gated paracellular channels and speculate that modulation of tight junction channel gating kinetics may be an unappreciated mechanism of barrier regulation.

*For correspondence: jrturner@rics.bwh.harvard.edu

Present address: [†]NDDD, Lupin Research Park, Pune, India

Competing interests: The authors declare that no competing interests exist.

## Introduction

Epithelial barriers are essential for the survival of multicellular organisms and allow compartmentalization and controlled interactions between distinct environments (*Marchiando et al., 2010*, *Turner, 2009*). While transcellular transport is mediated by proteins that span the plasma membrane, molecular details of ions, water, and solute transport across the tight junction, i.e. the paracellular path, are less well-defined. This, in part, reflects the absence of tools able to detect single channel events at the tight junction and, therefore, a reliance on methods that measure paracellular flux over large multicellular surfaces (*Shen et al., 2011*). The limited spatial and temporal resolution of these techniques has contributed to the widely held view of tight junction channels as constitutively open and has limited biophysical characterization of these paracellular channels. Nevertheless, it is clear that paracellular transport is critical to the function of transporting epithelia in many organs (*Bagnat et al., 2007*, *Simon et al., 1999*, *Wada et al., 2013*) and can be regulated by physiological and pathophysiological stimuli (*Heller et al., 2005*, *Marchiando et al., 2010*, *Suzuki et al., 2011*, *Turner et al., 1997*, *Weber et al., 2010*).

Intestinal epithelial expression of the tight junction protein claudin-2, which increases paracellular $Na^+$ conductance (*Amasheh et al., 2002*, *Wada et al., 2013*, *Weber et al., 2010*), is downregulated after the neonatal period (*Holmes et al., 2006*) but markedly upregulated in inflammatory and infectious enterocolitis and by several cytokines, including IL-13 (*Heller et al., 2005*). This is essential for

**eLife digest** Epithelial cells form layers that line the inner surface of the gut, lungs and other organs. They act as barriers to control the movement of water, ions and small molecules between internal compartments within the body and the external environment. Some substances are transported across these barriers by passing through individual epithelial cells, but others pass through the spaces between adjacent cells. These spaces are sealed by tight junctions. If the tight junctions do not work properly, it can cause problems with regulating the movement of molecules across epithelial-lined surfaces. This in turn can contribute to diseases in humans, including inflammatory bowel disease and chronic kidney disease.

Proteins called claudins form channels that only allow certain molecules to pass through tight junctions. One member of this family, called claudin-2, allows sodium ions and other small positively charged ions to cross between adjacent cells. However, it is not clear how these channels work, largely due to the absence of appropriate tools to study this process. Here, Weber et al. adapted a technique called patch clamping to study the behavior of individual claudin-2 channels in the tight junctions between mammalian epithelial cells.

Weber et al. found that claudin-2 allows positively charged ions to move across a tight junction in short bursts rather than in a steady stream as had been suggested by previous work. These bursts typically begin and end in less than a millisecond. Further experiments revealed that claudin-2 channels have several states; in one state the channel is fully open, in another the channel is firmly closed, and in the third state the channel is temporarily closed but primed to open.

Further experiments show that mutations in the gene that encodes claudin-2 or drugs that inhibit claudin-2's function alter the open and closed behaviors of these trans-tight junction channels. The technique developed by Weber et al. will enable researchers to understand how channel proteins at tight junctions assemble and operate. Such studies may lead to the development of drugs that can alter the activity of these channels to treat particular diseases.

IL-13-induced increases in paracellular $Na^+$ permeability, as conductance changes are prevented by siRNA-mediated inhibition of claudin-2 upregulation (*Weber et al., 2010*). Thus, claudin-2 expression is regulated during development and disease. Detailed functional analysis of claudin-2-based channels, in their own right and as models for all paracellular claudin channels, is therefore critical to understanding fundamental mechanisms of development and disease.

Claudin-2 expression induces large increases in paracellular flux of small cations (*Amasheh et al., 2002*, *Furuse et al., 2001*, *Weber et al., 2010*). Inducible claudin-2 expression in MDCKI monolayers, which lack endogenous claudin-2 expression, is therefore an ideal experimental model in which to define paracellular, trans-tight junction channels (*Angelow and Yu, 2009*). We analyzed claudin-2 channel function using a novel, trans-tight junction patch clamp technique. Here, we show that this approach can detect discreet conductance events, define these in biophysical terms, perform extensive characterization that demonstrates that the currents detected reflect the activity of trans-tight junction channels, and excludes the possibility that these events are due to apical membrane conductances or other artifacts. The data show that these tight junction channels are gated and behave in a manner reminiscent of traditional transmembrane ion channels despite radical differences in orientation and function. Nevertheless, these similarities, the efficacy of pharmacological effectors of transmembrane ion channel function, and the frequency of epithelial barrier defects in disease suggest that it may be possible to develop agents that positively- or negatively-regulate tight junction channel gating for therapeutic benefit.

## Results

As a reductionist system, we expressed claudin-2 under the control of a tet-off regulated system in polarized MDCKI monolayers (*Angelow and Yu, 2009*). Western blot demonstrated that, relative to the parental MDCKI line, small amounts of functional claudin-2 were present even when expression was repressed, i.e. doxycycline was present, consistent with the known minor leakiness of such regulated expression systems (*Figure 1A*). As expected based on previous comparisons of MDCKI and

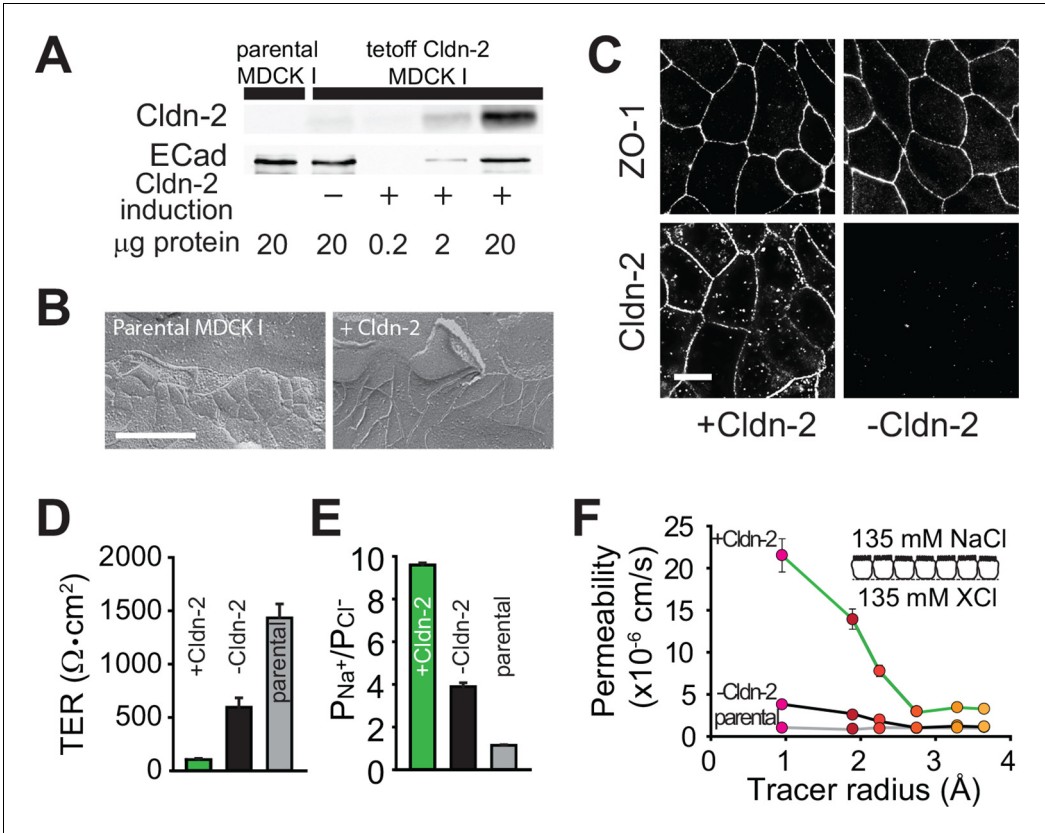

**Figure 1.** Claudin-2 expression enhances tight junction permeability to small cations. (A) Transgenic MDCKI monolayers were developed to express claudin-2 (+Cldn-2) in the absence of doxycycline. Limited claudin-2 expression was detected in the absence of induction and none was detectable in the parental MDCKI line. (B) Induction of claudin-2 expression had no effect on tight junction ultrastructure (Bar = 500 nm). (C) Tight junction claudin-2 was not detectible by immunofluorescence staining after suppression of claudin-2 expression (Bar = 10 μm). (D) Claudin-2 expression reduced TER (E) and increased relative permeability of sodium to chloride ($P_{Na^+}$/$P_{Cl^-}$) was increased. (F) Biionic potential analyses show that the reduction in TER was mainly due to increased paracellular permeability to small cations.

MDCKII cells (*Stevenson et al., 1988*), which differ primarily in their expression of claudin-2, induction of claudin-2 expression did not affect tight junction ultrastructure (*Figure 1B*). When expression was induced, claudin-2 concentrated at tight junctions and, to a limited extent, in cytoplasmic vesicles (*Figure 1C*). Induction of claudin-2 expression also reduced transepithelial electrical resistance (TER; *Figure 1D*) and increased cationic charge selectivity (*Figure 1E*) with strict size-selectivity (*Figure 1F*). Claudin-2 expression therefore induces charge- and size-selective increases in paracellular, trans-tight junction conductance.

## Claudin-2 expression induces conductance events that can be detected by trans-tight junction patch clamp

Measurements of paracellular permeability, such as those above, typically assess relatively large epithelial surfaces and, therefore, reflect global averages rather than local, site-specific conductances. Scanning and impedance approaches have been used in an effort to overcome these limitations (*Chen et al., 2013*, *Gitter et al., 1997*, *Krug et al., 2009*), but these lack the spatial and temporal resolution needed for identification of single channel events. Overall, the greatest obstacle to single channel analyses of tight junction channels has been the orientation of trans-tight junction channels between lateral surfaces of two adjacent cells, i.e. parallel to plasma membranes (*Figure 2A*). This orientation is orthogonal to traditional ion channels and gap junctions, which cross plasma

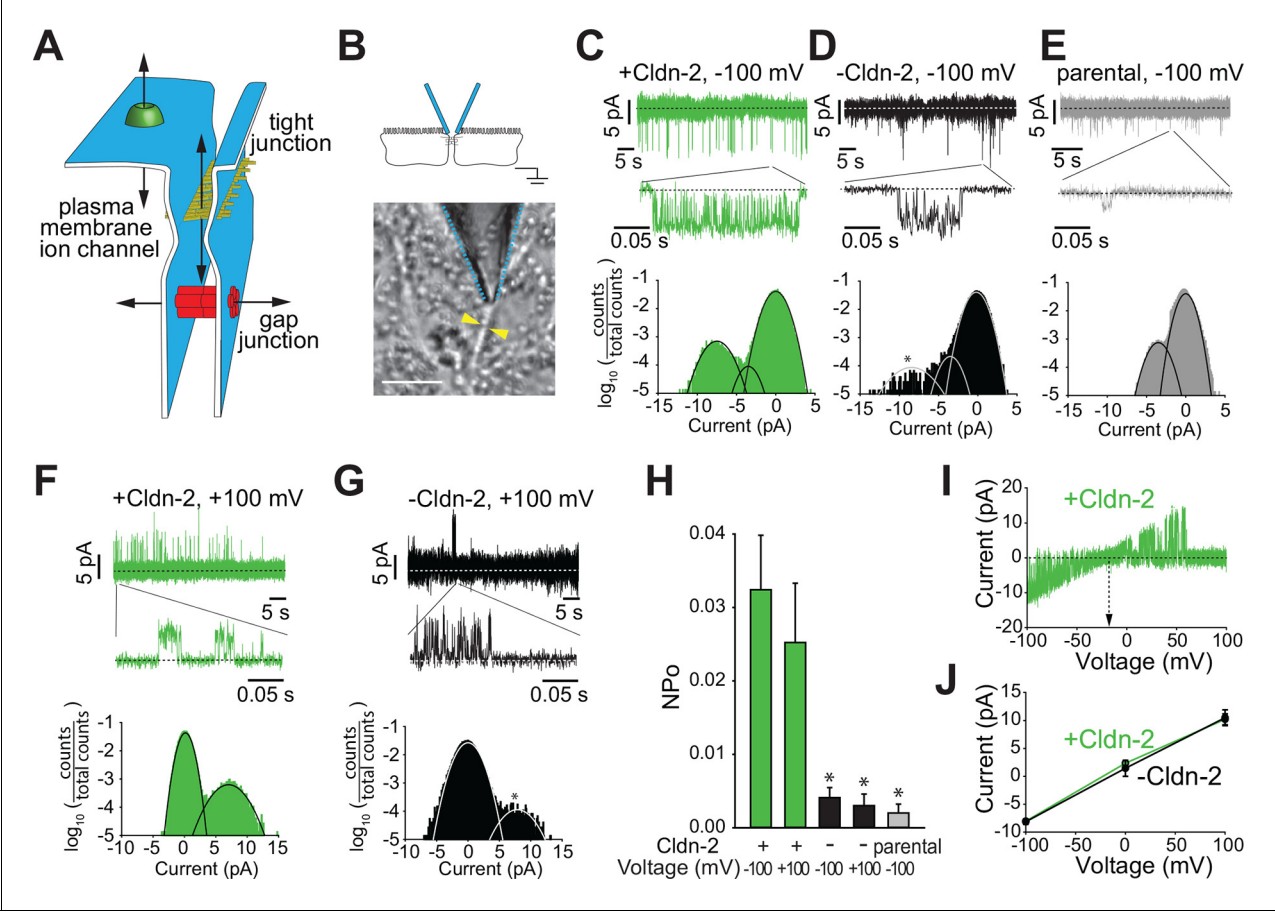

**Figure 2.** Claudin-2 expression correlates with the frequency of local tight junction channel openings in MDCKI monolayers. (**A**) Tight junctions are distinct from plasma membrane ion channels and differ from gap junctions in their ability to define conductance between two extracellular compartments. (**B**) Trans-tight junction patch clamp placement. Yellow arrowheads show intercellular junction (Bar = 10 µm). (**C**) Conductance events detected at −100 mV when claudin-2 was expressed (+Cldn-2). (**D**) In the absence of induced claudin-2 expression (–Cldn-2), the frequency of similar sized conductance events was dramatically reduced. (**E**) Small claudin-2 independent events were present in parental MDCKI monolayers (**F**) Conductance events were present at +100 mV when claudin-2 was expressed (+Cldn-2). (**G**) Events were infrequent in the absence of induced claudin-2 expression (–Cldn-2). (**H**) $NP_o$ was reduced by 87% ± 4% (at −100 mV) and 88% ± 6% (at +100 mV) after suppression of claudin-2 expression. Events were rare in recordings from parental tight junctions. (**I**) Representative recording of voltage ramp in claudin-2-expresing MDCKI monolayers showing linear current voltage relationship and reversal potential close to 0 mV. (**J**) Average current voltage relationships (n = 8 to 32 per condition) reveals that average channel conductance was ~90 pS regardless of whether claudin-2 expression was induced (green line) or not (black line).

membranes, and renders most patch clamp techniques unsuitable for measuring trans-tight junction ion flux.

To overcome these challenges, we developed an approach to seal an apical patch pipette across a region of the bicellular tight junction. This required several technical advances, including development of low profile chambers that allowed apical tight junction access using a 50°–60° approach angle while simultaneously viewing cell profiles from below the monolayer. Successful patching of tight junctions also required optimization of cell growth to afford clear morphological delineation of tight junctions while minimizing accumulation of cellular debris that could interfere with pipette sealing. Electrode configuration was also modified so that the pipette was just large enough to span the tight junction, while not so small that it would slip off of the tight junction. This allowed us to achieve a gigaseal with an ~5% success rate. Once a high resistance gigaseal was achieved, it was then possible to measure current through the paracellular pathway in response to an externally applied voltage relative to a basal reference electrode (*Figure 2B*).

The approach above allowed detection of bursts of sub-millisecond duration, flicker-like openings and closings when using holding potentials of −100 or +100 mV ($V_{apical}$− $V_{basal}$) in monolayers with

claudin-2 expression (*Figure 2C,F,H*). Such events were infrequent in monolayers without induction of claudin-2 expression, i.e. with low level claudin-2 expression (*Figure 2D,G,H*), and were rare in parental MDCKI monolayers that completely lacked claudin-2 expression (*Figure 2E,H*). All-points histograms, fitted to Gaussian distributions, show the claudin-2 dependent conductance centered at ~9 pA, and ranged from ~5 to >10 pA at –100 mV. A separate class of smaller conductance values centered at ~4.3 pA was present in all lines, regardless of claudin-2 expression.

To focus on claudin-2-dependent channels, thresholding was used to exclude the small, claudin-2-independent conductances. These analyses showed that opening probability ($NP_o$) of claudin-2-dependent channels was similar at +100 or –100 mV in claudin-2 expressing monolayers, but was reduced by 87 ± 4% (at -100 mV) and 88% ± 6% (at +100 mV) in the absence of claudin-2 induction (*Figure 2H*). In contrast, amplitude was similar at high and low levels of claudin-2 expression. Thus, the $NP_o$, but not the amplitude, of these openings with conductances of ~92 pS is a function of claudin-2 expression. This further suggests that the number of channels, but not the open probability of individual channels, is a function of claudin-2 expression.

To further characterize the voltage dependence of claudin-2-dependent conductances we performed voltage ramps beginning at a holding potential of –100 mV. Current-voltage (I-V) relationship plots (*Figure 2I,J*) showed that the reversal potential ($V_{rev}$) of these events was close to 0 mV, but slightly negative, and followed a linear function of voltage, consistent with a passive process. This result argues strongly that the conductance events cannot be apical cation or anion, e.g. $K^+$, $Na^+$, or $Cl^-$ channels, since the extracellular:intracellular gradients of these ions would necessitate equilibrium potentials much different than 0 mV. Notably, this analysis also shows that, despite there being far fewer events when claudin-2 expression was suppressed, individual conductance events were quantitatively similar, in both amplitude and duration, when claudin-2 expression was low (*Figure 2J*). Therefore, the ~9 pA single channel conductances measured by trans-tight junction patch clamp are non-vectorial, as expected for passive paracellular channels, and is unlikely be due to the activity of apical transmembrane ion channels.

## Kinetic analysis shows that claudin-2 channels have one open and two closed states

The observation that claudin-2-dependent conductance events occur in bursts was somewhat surprising, as tight junction conductance has been assumed to be uniform over time. This widely-held belief was based on stable measurements of paracellular conductance across large epithelial surfaces. However, conductance of individual channels is averaged over space in these measurements, which lack both temporal and spatial resolution of the trans-tight junction patch clamp approach.

To better characterize the opening and closing behaviors of claudin-2-dependent channels, histograms of all events were generated. Opening duration histograms of tight junction patch clamp data from cells with high or low claudin-2 expression at holding potentials of +100 or -100 mV. These revealed a single population of openings with $\tau_{open} < 1$ ms (*Figure 3A–D*). In contrast, closed duration histograms under the same conditions revealed two populations of closings, corresponding to closed states between and within event clusters. Interburst closings were prolonged with $\tau_{closed(stable)} > 1$ s while intraburst closings occurred with millisecond kinetics, i.e. $\tau_{closed(transient)} < 2$ ms (*Figure 3E–H*). One possibility is that the prolonged state (closed$_{stable}$) could represent channel disassembly, while the shorter closed state (closed$_{transient}$) results from regulation of assembled channels. However, given the absence of a significant vesicular claudin-2 pool and the relatively long ~9 hr half-life of claudin-2 protein in MDCK monolayers (*Van Itallie et al., 2004*), we consider it highly unlikely that vesicular traffic or protein turnover could be responsible for the observed opening and closing events. In contrast, although the pool of claudin-2 at the tight junction is largely immobile (*Raleigh et al., 2011*), the limited intramembranous diffusion that does occur (*Shen et al., 2008*) has kinetics within the range of the longer closed state (closed$_{stable}$) and we cannot exclude this as a possible regulatory mechanism.

Similar to amplitude, open and closed state kinetics were similar under high or low claudin-2 expression. Claudin-2 channel gating is thus independent of expression level, i.e. is non-cooperative. This also indicates that changes in $NP_o$ that occur as a function of claudin-2 expression reflect differences in channel number rather than open probability of individual channels. Further, because properties were similar at +100 and -100 mV, we can conclude that claudin-2-dependent channels are not gated by voltage, unlike many transmembrane ion channels. Overall these data show that

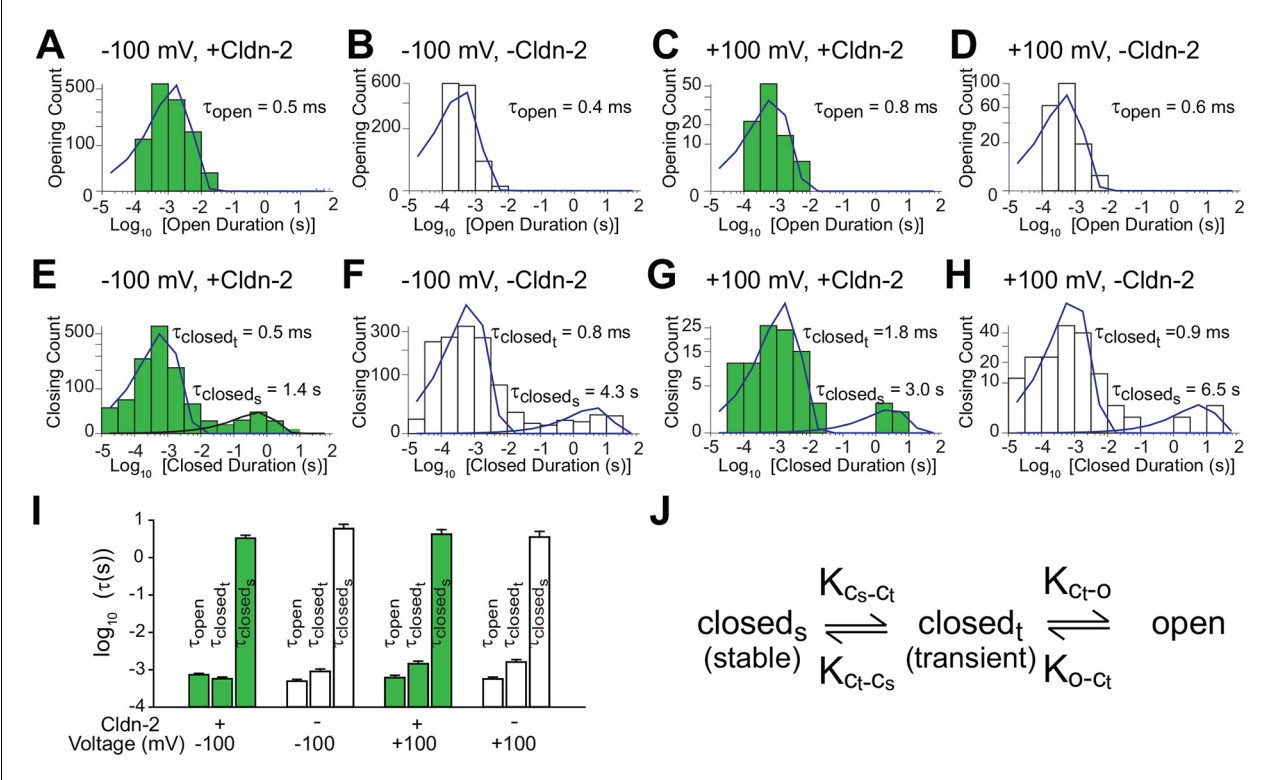

**Figure 3.** Patch clamp recordings reveal a single open state and two closed states. (A–D) A single population of fast openings was observed in the presence (green) or absence (white) of induced claudin-2 expression at –100 and +100 mV (a-d total recording times: 40 s, 225 s, 31 s, 52 s.). (E–H) Corresponding closed duration histograms from the same representative recordings reveal two distinct closed states. (I) Opening and closing time constants were voltage independent and were similar with and without claudin-2 induction (n=7 to 35 recordings for each condition). (J) Kinetic analysis demonstrates the presence of both stable ($c_{stable}$) and transient ($c_{transient}$) closed states and one and open (o) state.

claudin-2-dependent channels can exist in a highly dynamic opening state (o) as well as stable ($c_{stable}$) and transient ($c_{transient}$) closed states (*Figure 3J*).

## Claudin-2-dependent openings detected by trans-tight junction patch clamp are resistant to traditional ion channel inhibitors

Despite the voltage ramp results (*Figure 2I,J*), we re-considered the possibility that detected events represented transmembrane conductances of apical ion channels within apical membrane captured by the patch pipette. The ~9 pA claudin-2-dependent openings were, however, never observed when electrodes were sealed off of the tight junction, i.e. over apical membranes away from the tight junction. In place of the ~9 pA openings, small conductances could sometimes be detected when electrodes were sealed over apical membranes, but only when the data were low-pass filtered at 500 Hz (*Figure 4A*). These events differed distinctly from the claudin-2-dependent conducances, as the former were more common at holding potentials of +100 mV, relative to -100 mV, and had amplitudes of less than 2 pA, well below those of claudin-2-dependent events. These data provide spatial evidence that the conductances detected by trans-tight junction patch clamp are not traditional, transmembrane apical ion channels.

The data above, including the gigaohm seals achieved, symmetrical behavior, near 0 mV reversal potential, detection only at tight junctions, and claudin-2-dependence, suggest that the ~9 pA events detected by trans-tight junction patch clamp cannot be due to artifacts, such as transmembrane ion channels in apical membrane domains sealed within the patch pipette or pipette leak. We nevertheless took a pharmacological approach to further examine the hypothesis that these conductance events represented activity of transmembrane ion channels. Three different inhibitor cocktails were added to the patch pipette (*Figure 4B*). The first cocktail contained 10 mM 4-aminopyridine

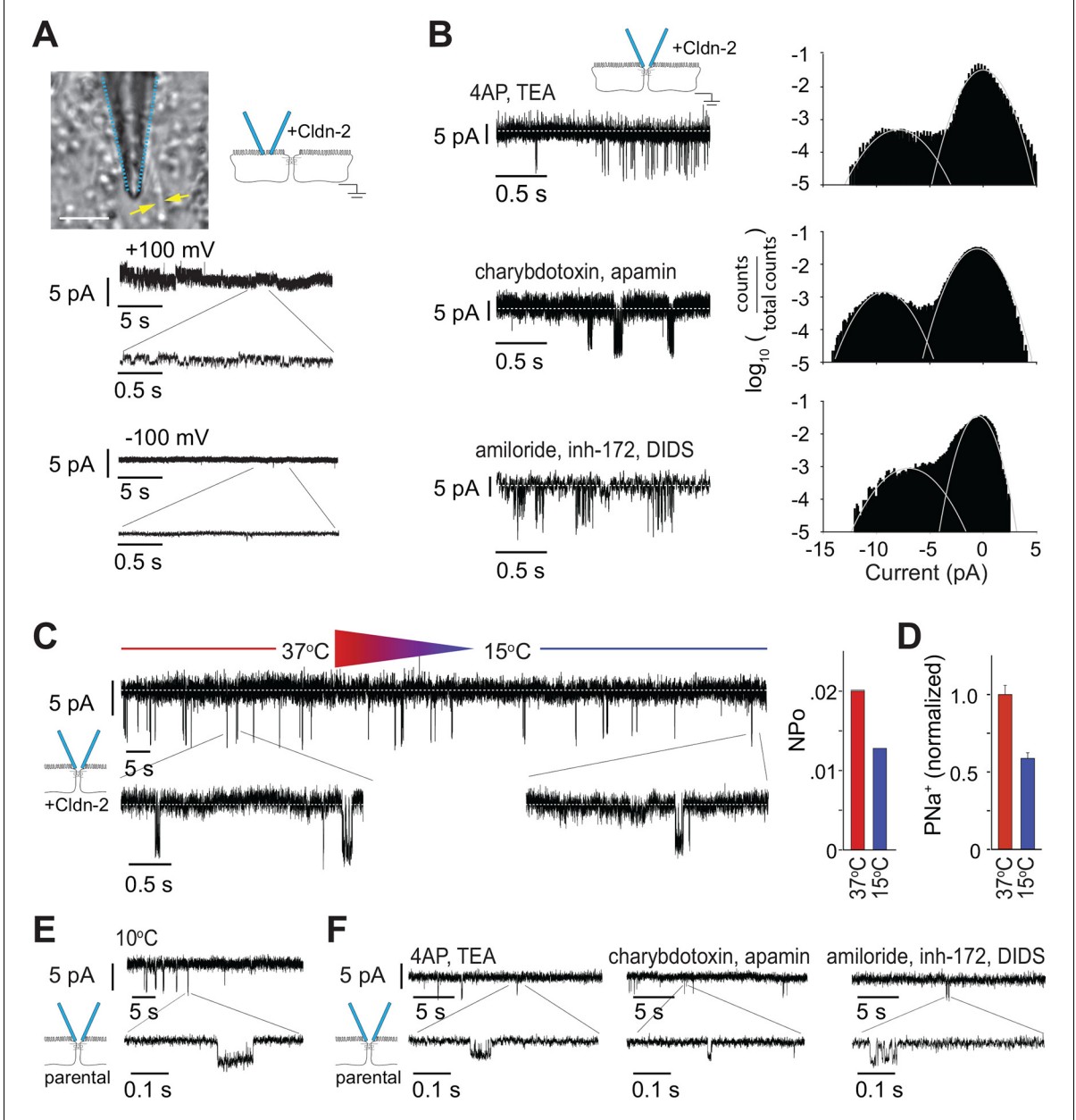

**Figure 4.** Large and small tight junction currents are not due to transmembrane ion channels. (A) Small (<2 pA) transmembrane ion channel openings were detectable in off-tight junction recordings after applying a 500 Hz low pass filter. (Bar = 10 μm). (B) Events detected by trans-tight junction patch clamp were not blocked by three different ion channel inhibitor cocktails (+Cldn-2; representative of n = 3 to 8 per condition). (C) Monolayers were cooled while recording from trans-tight junction patch clamp. The number of events detected was reduced at 15°C, relative to 37°C, but event amplitude was unaffected (+Cldn-2; representative of n = 4). (D) $NP_o$ and $Na^+$ permeability measured across a 0.33 cm$^2$ monolayer were similarly reduced at 15°C (+Cldn-2). (E) ~4 pA events remained detectable after chilling monolayers to 10°C (MDCKI parental monolayers; representative of n = 4). (F) ~4 pA events were not blocked by three different ion channel inhibitor cocktails (MDCKI parental monolayers; representative of n = 3 to 5 per condition).

and 10 mM TEA-Cl to inhibit voltage-activated $K^+$ channels. The second cocktail contained 100 nM charybdotoxin and 2 μM apamin to inhibit small conductance $Ca^{2+}$ activated $K^+$ channels. The third cocktail contained 100 μM amiloride, 20 μM CFTR inhibitor-172 (inh-172), and 250 μM 4,4'-Diisothio-cyano-2,2'-stilbenedisulfonic acid (DIDS) to block epithelial $Na^+$ channels, CFTR, and anion exchangers respectively. None of these affected frequency or amplitude of the claudin-2-dependent ~9 pA

events detected by trans-tight junction patch clamp. We therefore conclude, on the basis of biophysical, molecular, and pharmacological data, that the ~9 pA events detected represent single-channel conductances across the tight junction.

## Temperature sensitivity of claudin-2 channels is similar whether measured by trans-tight junction patch clamp or traditional, global approaches

Previous studies have shown that overall transepithelial conductance as well as claudin-2-dependent $Na^+$ conductance decrease when temperature is reduced (*Gonzalez-Mariscal et al., 1984*, *Martinez-Palomo et al., 1980*, *Shen et al., 2008*, *Yu et al., 2009*). We therefore assessed temperature sensitivity of the single channel events measured by trans-tight junction patch clamp. These studies were complicated by the technical challenge of cooling the monolayer without creating excessive electrical noise that prevented analysis and limited cooling to ~20°C while recording from the trans-tight junction patch clamp. When monolayers of claudin-2-expressing MDCKI were cooled from 37°C to 15°C while recording, the number of claudin-2-dependent (~9 pA) events fell by 37% (*Figure 4C*). This was closely paralleled by a 41% decrease in $Na^+$ permeability measured across monolayers using traditional approaches (*Figure 4D*), providing more support for the conclusion that the ~9 pA conductance events reflect activity of paracellular claudin-2 channels.

We also assessed the claudin-2-independent, ~4 pA events using the parental MDCKI cells that completely lacked claudin-2 expression. We did this because these claudin-2-independent currents can be obscured by larger events (e.g. *Figure 2*). In contrast to the ~9 pA events, the ~4 pA events were resistant to cold (p = 0.35), even when chilled to 10°C (*Figure 4E*). The ~4 pA events were also resistant to all of the ion channel inhibitor cocktails (*Figure 4F*).

## Claudin-2 channel behavior is consistent across different types of epithelia

The MDCK cell line is derived from epithelia of the distal convoluted tubule, which function effectively to absorb $Na^+$ and $Cl^-$ in the apical-to-basal direction and secrete $K^+$ (*Gekle et al., 1994*). Cell lines derived from epithelia within other parts of the body have distinct specialized functions. For example, Caco-2 cells are a human colon epithelial cancer cell line that differentiate as absorptive enterocytes and express brush border enzymes and transporters typical of this cell type (*Peterson and Mooseker, 1992*, *Pinto et al., 1983*, *Turner and Black, 2001*, *Turner et al., 1996*, *Turner et al., 1997*). While both MDCK and Caco-2 cell lines are both commonly used to study polarized epithelial cell function, their distinct phenotypes are reflected by divergence in both function and protein expression. Nevertheless, tight junction ultrastructure is similar in MDCK and Caco-2 cells, and both are composed of three to five strands (*Figure 5A*), which express claudin-2 abundantly (*Figure 5B,C*). We took advantage the availability of a Caco-2$_{BBe}$ line in which claudin-2 expression was stably knocked down (*Raleigh et al., 2011*) to assess the effects of claudin-2 depletion, rather than addition, on paracellular channel function. This also allowed direct comparison of canine renal epithelia and human intestinal epithelia.

Paracellular $Na^+$ conductance across Caco-2$_{BBe}$ monolayers was similar to that of MDCKI monolayers with induced claudin-2 expression (*Figure 5D* vs *Figure 1F*). When claudin-2 expression was stably suppressed by shRNA-mediated knockdown, $Na^+$ conductance across Caco-2$_{BBe}$ monolayers was greatly diminished (*Figure 5D*) and fell to a level similar to the claudin-2-deficient MDCKI parental line (*Figure 1F*).

We next analyzed monolayers of Caco-2$_{BBe}$ human intestinal epithelia by trans-tight junction patch clamp (*Figure 5E*). It was substantially more difficult to achieve a gigaohm seal in these monolayers relative to MDCK, likely due to the well-developed brush border of Caco-2$_{BBe}$ cells. As with MDCK monolayers, all points histograms demonstrate conductance values centered around ~8 pA, i.e. ~82 pS, in claudin-2-expressing monolayers (*Figure 5F*), and there was a specific reduction in this class of events after claudin-2 knockdown (*Figure 5G*). We therefore conclude that claudin-2 depletion in human intestinal epithelia eliminates events similar to those generated by claudin-2 expression in canine renal epithelia.

While $NP_o$ was reduced, conductance of the claudin-2-dependent events was not affected by claudin-2-knockdown (*Figure 5H*), demonstrating that single channel conductance was not a

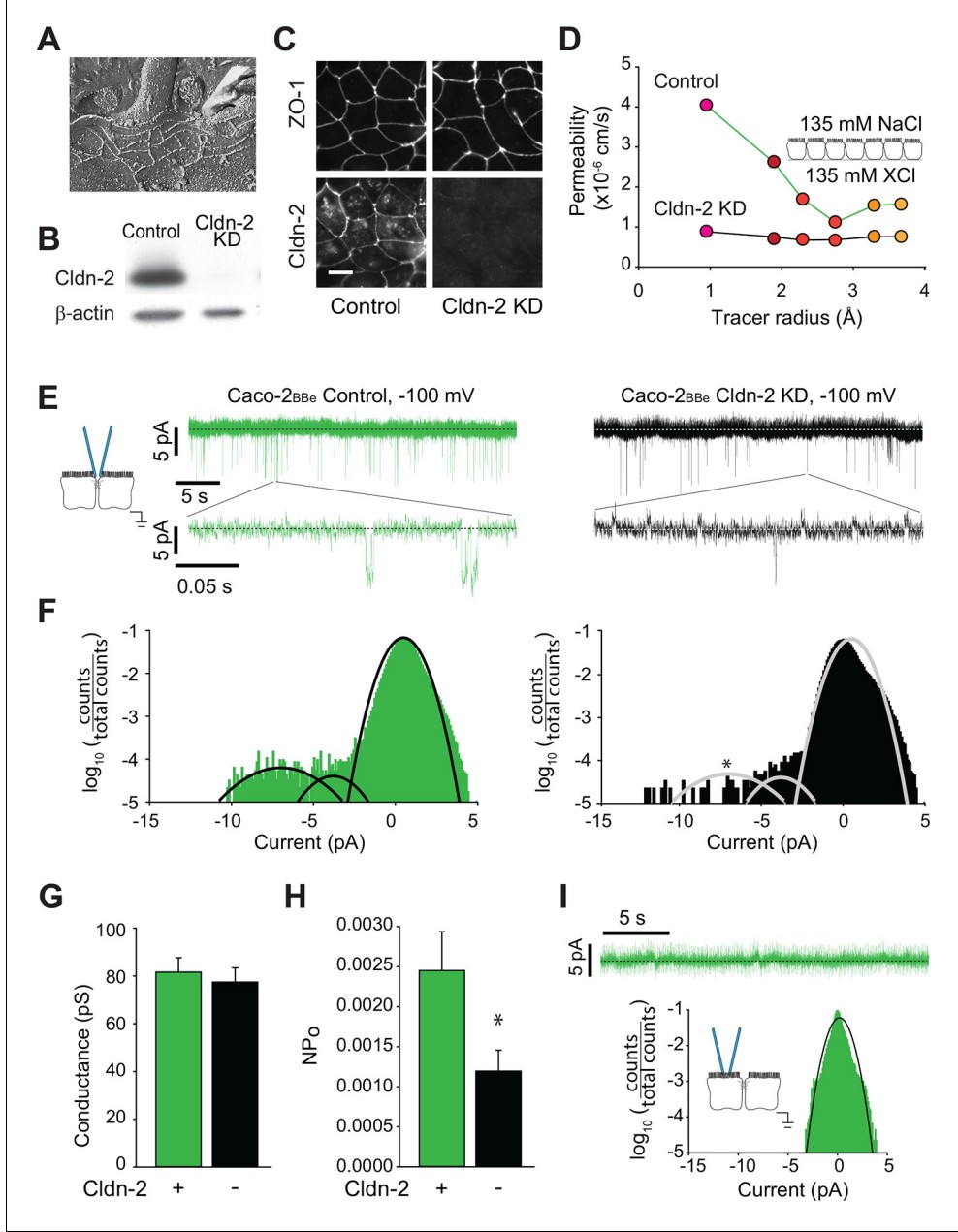

**Figure 5.** Global conductance and trans-tight junction patch clamp event frequency correlate with claudin-2 expression in Caco-2$_{BBe}$ intestinal epithelial monolayers. (**A**) Freeze fracture electron microscopy demonstrating that mature tight junctions in Caco-2$_{BBe}$ monolayers are composed of 3–5 strands (***Shen et al., 2006***), similar to MDCKI. (**B**) Western blot confirms >99% knockdown of claudin-2 in Caco-2$_{BBe}$ monolayers. (**C**) Claudin-2 is not detectable by immunofluorescence microscopy of knockdown Caco-2$_{BBe}$ monolayers (Bar = 10 μm). (**D**) Biionic potential analyses show that claudin-2 knockdown reduces small cation permeability. (**E**) Trans-tight junction patch clamp recordings of Caco-2$_{BBe}$ cells detected events at −100 mV (n=5 per condition). Representative traces of trans-tight junction patch clamp data from control and claudin-2 knockdown Caco-2$_{BBe}$ monolayers. (**F**) All points histogram analysis of patch clamp data from Caco-2$_{BBe}$ monolayers shows a specific reduction in ~9 pA events with no change in frequency of ~4 pA events after claudin-2 knockdown. (**G**) Average opening conductances was unaffected by the levels of claudin-2 expression. (**H**) Channel activity (NP$_{o}$) was reduced by claudin-2 knockdown (n = 5 to 7 per condition). (**I**) Neither ~9 pA nor ~4 pA events were not detectable when the pipette was sealed away from the tight junction in Caco-2$_{BBe}$ monolayers (n = 12).

function of claudin-2 concentration. Similar to the data from MDCK monolayers, this indicates that gating of claudin-2-dependent channels is not cooperative. A class of smaller conductances that were not affected by claudin-2 knockdown was also detected, similar to the claudin-2-independent events detected in MDCK monolayers. Finally, as in MDCK cells, the claudin-2-dependent openings were not detectable when the patch pipette was sealed away from the tight junction in Caco-2$_{BBe}$ monolayers (*Figure 5I*).

Thus, these data demonstrate the tight junction opening events are detectable in two very different epithelia derived from different organ systems, and, in both cases, the events were claudin-2 dependent. Further, these data exclude the possibility that events could be due to off-target effects induced by the tet transactivator used to drive claudin-2 expression in MDCK cells. More importantly, the similar conductance values of these events, despite markedly different NP$_o$ values, supports the conclusion that the events are mediated by biophysically similar claudin-2-based paracellular channels, rather than some other paracellular channel that might be expected to differ between canine renal and human intestinal epithelia.

## Claudin-2-dependent conductances are driven by transepithelial ion gradients and are charge- and size-selective

Paracellular flux across tight junction channels is driven passively by transepithelial electrochemical gradients. To determine if flux through the claudin-2 dependent channels detected by trans-tight junction patch clamp behaves similarly, we measured single channel reversal potentials in the presence of large apical:basolateral or basolateral:apical NaCl gradients. As predicted for a paracellular conductive pathway, we observed similar shifts in reversal potential but in opposite directions by iso-osmotically replacing 90% of basolateral or apical NaCl with mannitol (*Figure 6A,B*). Importantly, such symmetrical behavior indicates that flux across these channels is electrochemically driven by apical:basolateral gradients. In contrast, transmembrane ion channels are driven by extracellular:intracellular gradients. As intracellular cation composition does not change as rapidly as the extracellular media, transmembrane ion channels would not be expected to behave symmetrically under these conditions.

A second very important feature of the reversal potential measurement approach is that permeation occurs via a passive, yet selective, mechanism defined by the electrochemical gradients of the ions passing through the channel. This allowed us to further characterize the charge-selectivity of claudin-2-dependent conductances. The magnitude of reversal potential shifts indicates that the P$_{Na}^+$/P$_{Cl}^-$, as defined by Nernst equilibrium potentials, of single channel events detected by trans-tight junction patch clamp is 7.4 ± 3.7. This is similar to the P$_{Na}^+$/P$_{Cl}^-$ of 9.5 ± 0.1 measured across intact monolayers using traditional global measurement approaches (*Figure 1D*). In addition to demonstrating that the detected single channel events have P$_{Na}^+$/P$_{Cl}^-$ that is indistinguishable from that of tight junctions, these data also exclude the possibility that openings represent anion channels, such as CFTR, that would be expected to be Cl$^-$, rather than Na$^+$, selective.

When measured across intact monolayers, paracellular tight junction permeability is well recognized to be size-selective. This can be assessed by measuring flux of differently-sized uncharged probes. However, the temporal averaging required by these approaches precludes detection of transient, single-channel events. Fortunately, this obstacle has recently been overcome using the technique of biionic substitution, where Na$^+$ within the basolateral media is replaced with a larger cation (*Angelow and Yu, 2009*, *Shen et al., 2011*). Use of the trans-tight junction patch clamp in conjunction with biionic substitution showed that the channels were permeable to methylamine (radius=1.9 Å) and that, like Na$^+$, the V$_{rev}$ of methylamine was close to 0 mV (*Figure 6C*). This indicates that conductance of Na$^+$ and methylamine is similar in magnitude. In contrast, channels detected by trans-tight junction patch clamp were relatively impermeant to the larger cations tetramethylammonium (radius=2.8 Å) and N-methyl-D-glucamine (radius=3.6 Å). Thus, basolateral Na$^+$ replacement with these larger cations induced a large negative shift in channel reversal (*Figure 6D–F*). Global analyses of claudin-2-expressing MDCK by biionic substitution demonstrated tight junction size-selectivity that paralleled that of conductance events detected by trans-tight junction patch clamp (*Figure 6G*). In addition to providing another biophysical measure in which the claudin-2-dependent events detected by trans-tight junction patch clamp are similar to claudin-2-dependent paracellular conductances measured by traditional averaging approaches, these data provide further support for the conclusion that the events detected cannot represent apical K$^+$ channels, as the

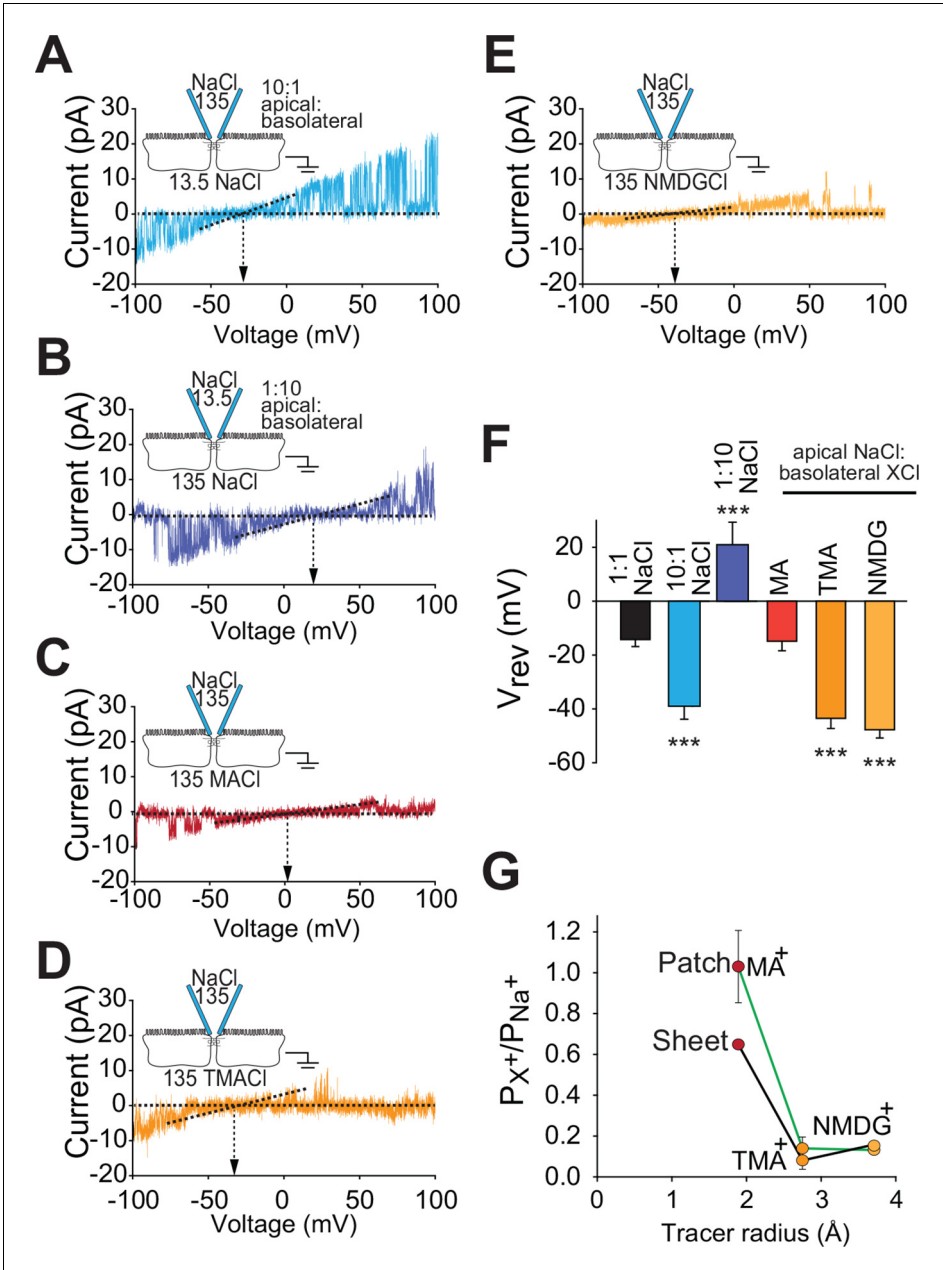

**Figure 6.** Claudin-2-dependent conductance events measured by trans-tight junction patch clamp display cation- and size-selective properties similar to transepithelial paracellular conductance measured over large areas. (A–E) Current-voltage (I-V) relationships for events detected by trans-tight junction patch clamp in MDCKI monolayers with transgenic claudin-2 expression (+Cldn-2) under the ionic conditions shown. Pipette and basolateral buffer composition are indicated in mM. (F) $V_{rev}$ was determined under each of the ionic conditions shown (n=4 to 7 per condition). (G) Cation permeability determined from shifts in trans-tight junction patch clamp $V_{rev}$ (green line) or traditional (black line) bi-ionic potential measurements.

latter are highly-selective and do not accommodate cations as large as methylamine. It is, however, interesting to note that the permeability of methylamine, relative to $Na^+$, is greater in patch clamp measurements than in global measurements. We do not know the reason for this, but could speculate that it may be an artifact of the physical forces created by sealing the patch pipette over the junction. However, even if that explanation was correct, the fact that permeabilities of tetramethylamine and N-methyl-D-glucamine, relative to $Na^+$, are unaffected and that charge selectivity is

maintained show that such effects, if present, are limited. These data therefore demonstrate that the claudin-2 dependent channel represents a passive paracellular conductance pathway with charge- and size-selectivities similar to those measured across large epithelial sheets.

## Tight junction conductances are blocked by non-specific and specific inhibitors of claudin-2

$La^{3+}$ is known to nonspecifically inhibit claudin-2-dependent paracellular flux (*Machen et al., 1972*, *Yu et al., 2010*). Consistent with this, addition of $La^{3+}$ to the basolateral chamber completely blocked claudin-2 dependent opening events (*Figure 7A–C*). These data, along with the ionic substitution experiments above, indicate that the channels being studied are equally accessible from apical and basolateral approaches and again refute hypotheses suggesting that the openings detected represent plasma membrane ion channels.

$La^{3+}$ is, however, a non-selective inhibitor. We therefore sought a more specific approach to block claudin-2 channels. Recent work has begun to define the structural basis for ion selectivity of paracellular, claudin-dependent, trans-tight junction channels (*Angelow and Yu, 2009*, *Li et al., 2013*, *Li et al., 2014*, *Li et al., 2013*, *Yu et al., 2009*). Extracellular loop 1 (ECL1) of claudin-15 forms the first four β strands, and charged residues at the end of the fourth β strand are thought to line the claudin channel and serve as critical determinants of charge selectivity (*Angelow and Yu, 2009*, *Colegio et al., 2003*, *Suzuki et al., 2014*). Within the corresponding region of claudin-2, $Ile^{66}$ is thought to be buried within a narrow part of the channel (*Angelow and Yu, 2009*). Covalent modification of claudin-$2^{I66C}$ using 2-(trimethylammonium) ethyl methanethiosulfonate (MTSET) reduced global paracellular cation flux in claudin-$2^{I66C}$, but not claudin-$2^{WT}$, consistent with obstruction of the claudin-2 pore (*Figure 7D*).

We exploited the observation that claudin-$2^{I66C}$ can be inhibited by MTSET to ask whether the single channel conductances can be similarly inhibited using MDCKI monolayers with inducible expression of claudin-$2^{I66C}$ (*Figure 7E*). Trans-tight junction patch clamp recordings demonstrated that MTSET markedly reduced $NP_o$ of MDCKI monolayers expressing claudin-$2^{I66C}$ (*Figure 7F,H,J*) with kinetics similar to MTSET inhibition measured by global approaches (*Angelow and Yu, 2009*). As expected, MTSET had no effect on channel events in monolayers expressing claudin-$2^{WT}$(*Figure 7G,I*). In most cases, MTSET completely abolished events detected by trans-tight junction patch clamp in monolayers expressing claudin-$2^{I66C}$ (*Figure 7H*), but a few residual events persisted in some monolayers (*Figure 7H*). These MTSET-resistant events were identical in size to those observed prior to MTSET addition (*Figure 7J*), suggesting that MTSET inhibits each claudin-2 channel either completely or not at all. The small number of MTSET-resistant channels occurred in a subset of monolayers, suggesting that this may reflect the lability of MTSET in aqueous solutions. Alternatively, the observation that events detected in MTSET-treated claudin-$2^{I66C}$-expressing monolayers had conductances of ~92 pS, similar to monolayers expressing claudin-$2^{WT}$ could be interpreted as support for a role of $Ile^{66}$ in claudin-2 gating. In either case, these data show that site-directed mutagenesis and chemical derivatization of $Ile^{66}$ blocks local conductance events in a specific manner. Because claudin-2 is concentrated at the tight junction and has not been demonstrated to create transmembrane channels, this result provides further support for the conclusion that events represent flux across paracellular, i.e. tight junction, channels.

## Discussion

We have defined the biophysical behavior of claudin-2-dependent single channel conductance events in both canine kidney and human intestinal epithelia. The data confirm that claudin-2 channels within tight junctions are driven passively by the transepithelial, apical to basolateral electrochemical gradient. Analyses of single channel events show that the single channel size- and charge-selectivity of these channels is nearly identical to values obtained when the same parameters are assessed across large epithelial sheets using traditional methods. Further, single channel events were inhibited by $La^{3+}$, reduced temperature, and, more specifically, cysteine mutagenesis and chemical derivitization of $Ile^{66}$ within the claudin-2 pore. In contrast, attempts to block single claudin-2 openings with numerous established transmembrane ion channel blockers were unsuccessful. Moreover, the symmetrical behavior and size-selectivity of the claudin-2-dependent events detected by trans-tight junction patch clamp are inconsistent with known and predicted transmembrane, e.g. apical or

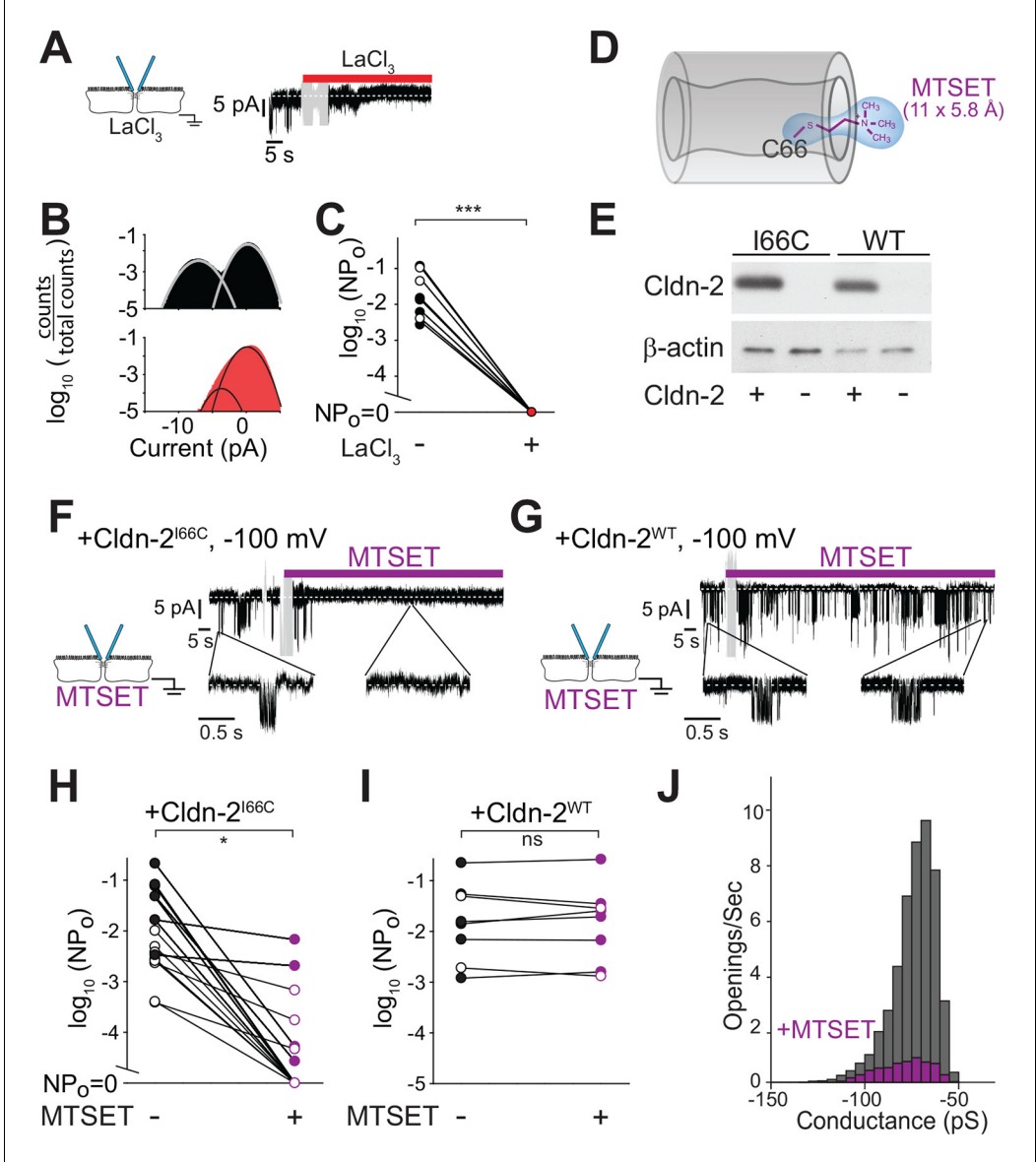

**Figure 7.** Paracellular conductance events are blocked by La$^{3+}$ or claudin-2 derivatization. (A) LaCl$_3$ (red bar) blocked opening events. Solution exchange artifact is shown in gray. (B) The ~9 pA events were eliminated from the all-points histogram by La$^{3+}$ treatment (before LaCl$_3$ black; after LaCl$_3$ red) (C) La$^{3+}$ treatment (red) reduced NP$_o$ to 0 (closed symbols indicate measurements at -100 mV, open symbols indicate measurements at +100 mV, n = 9). (D) MTSET forms a disulfide bond with Cys$^{66}$ located within the pore of claudin-2$^{I66C}$ channels (**Angelow and Yu, 2009**). (E) Transgenic claudin-2$^{I66C}$ was expressed at levels similar to claudin-2$^{WT}$. (F,G) MTSET dramatically reduced the number of detectable events in trans-tight junction patch clamp recordings from MDCKI cells expressing claudin-2$^{I66C}$ within ~20 s, but had no effect on monolayers expressing claudin-2$^{WT}$. Blue bar indicates presence of MTSET (n = 8 to 16 per condition). Solution exchange artifacts are shown in gray. (H,I) NP$_o$ of MDCKI cells expressing claudin-2$^{I66C}$ or claudin-2$^{WT}$ before and after (purple) MTSET treatment (closed symbols indicate measurements at -100 mV, open symbols indicate measurements at +100 mV, n = 8 to 16 per condition). (J) Derivatization of claudin-2$^{I66C}$ does to affect conductance of residual events (V = –100 mV). The histogram depicts frequency of events before and after (purple) MTSET treatment.

basolateral membrane, ion channels. Thus, we must conclude that the conductance events detected do represent opening and closing of trans-tight junction channels.

It is, perhaps, surprising that trans-tight junction flux occurs via highly dynamic, rather than static, paracellular channels that are gated with sub-millisecond kinetics. Nevertheless, the data clearly show that this is the case. This suggests that, despite being oriented parallel to, rather than across, the plasma membrane, paracellular channels open and close in a manner similar to typical ion

channels that span the plasma membrane. Claudins do not, however, have any sequence similarity to known transmembrane ion channels. Thus, claudin-dependent tight junction channels represent an entirely new class of dynamically gated ion conductance pathways.

## Structural basis of tight junction conductive pathways

Tight junction ultrastructure, as seen using freeze-fracture electron microscopy, is established by an anastomosing arrangement of strands which encircle the cell at the apical intercellular space. It has been proposed that these strands establish the paracellular barrier and that the strands are also populated by "channels" which impart charge and size selectivity. Whether these channels, commonly referred to as the pore pathway (*Shen et al., 2011*), are open at steady state or can open and close was previously unknown. Our data show that tight junctions are populated by highly dynamic, gated channels that transition between at least three different states. The means by which these functional state transitions occur and whether they are or can be regulated is one interesting question that follows from the results presented here. Further, as investigators in this area seek to model claudin channel function based on the published crystal structure of claudin-15 (*Suzuki et al., 2014*), our data indicate that any model applied to claudin-2 must include an explanation for rapid opening and closing of the channel. One could hypothesize that, in contrast to the shorter closed state, the longer closed state may reflect transient disassembly of the claudin-2 channel complex.

We speculate that some of the properties observed in our single channel recordings are due to the arrangement of channels spanning tight junction strands that are oriented in both series and parallel, implying that multiple simultaneous tight junction channel openings would result in stepwise conductance increases. Indeed, we did observe short-lived, step-wise conductance increases, consistent with a parallel opening of a second claudin-2 channel, superimposed on a typical initial opening (e.g., *Figures 4B,6B,E* and *7G*). Alternatively, the multi-strand tight junction ultrastructure and corresponding conductance model proposed by Claude (*Claude, 1978*) suggest that channels are also arranged in series. If true, this would indicate that the $NP_o$ measured here for current across the entire height of the tight junction significantly underestimates the $NP_o$ of individual claudin-2 channels. Consistent with this, we did observed some variability in claudin-2-dependent current amplitude. This could be related to the specific configuration of a claudin-2 channel within a given submicron segment of tight junction. For example, heterogeneity may be due to variations in baseline strand conductance, different numbers of tight junction strands, or variations in strand branch points within a particular patch of tight junction. Indeed, the anastamosing arrangement of tight junction strands may be a mechanism that limits lateral spread of current from single channel openings within any individual strand.

## Electrical analysis of trans-epithelial conductive pathways

To better understand the different types of events detected within the patch clamp recordings presented here, we developed an equivalent circuit diagram (*Figure 8*). First, we determined resistance of the patch pipette electrode itself. When in solution, i.e. free of cells, the resistance was determined to be 2.5 MΩ, similar to that of electrode used by others. When the pipette was sealed to the monolayer, resistance of the seal leak, i.e. the path resulting from incomplete electrical isolation at the edges of the pipette, was determined to be ~30 GΩ, based on the steady-state current of (~3.5 pA) at -100 mV when the pipette was sealed over the junction of claudin-2-deficient monolayers composed of either parental or uninduced, transfected MDCKI cells. The resistance of the paracellular shunt pathway (outside of the pipette) is simply the measured resistance of the monolayer, which ranged from 100 to 1,500 Ω·cm², i.e. 0.0003 to 0.0045 MΩ. Because this resistance is so much lower than any other pathway measured, it is essentially 0 for these analyses, as it cannot significantly impact conductance events of the types measured here. Thus, after accounting for low pipette seal leak and electrode resistance, the conductance events detected by the patch pipette can only represent transepithelial currents.

To specifically assess non-tight junction conductance pathways that could, for example, be due to apical membrane trapped within the patch pipette, the pipette was sealed over the apical membrane, i.e. away from the junction. In this configuration, ~2 pA currents could be detected at 100 mV, which corresponds to a pathway with a resistance of 50 GΩ. Two additional pathways, measuring ~9 pA and ~4 pA events were present only when the pipette was sealed over the junction. The

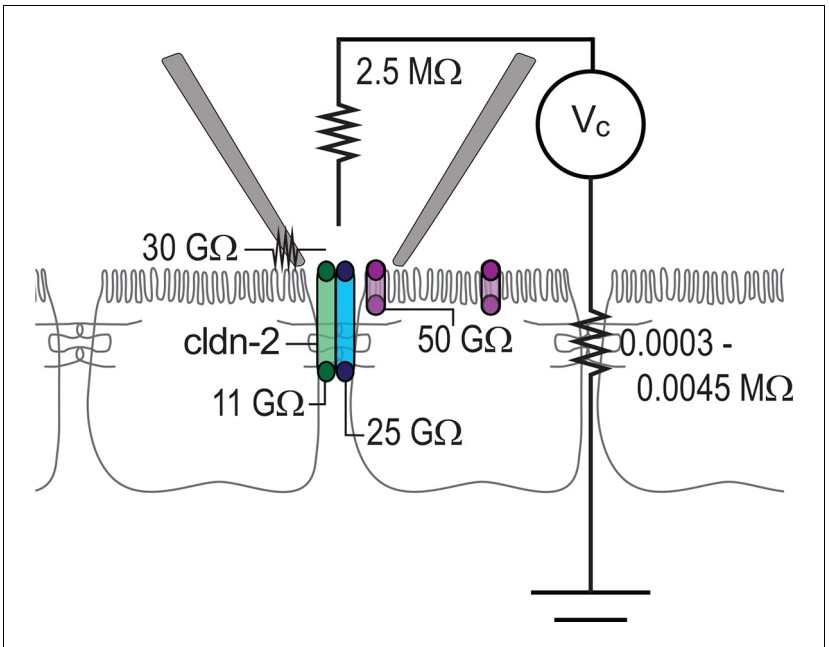

**Figure 8.** Circuit analysis of current pathways detected by trans-tight junction patch clamp. Resistances of the pipette seal, the electrode, the paracellular pathway of the larger epithelial sheet (outside of the patch), detectable apical membrane channels, and both claudin-2-dependent (green) and claudin-2-independent (blue) paracellular channels are shown. Both paracellular channels were detected only when the patch pipette was sealed over the intercellular junction. The apical membrane channels were only detected when the patch pipette was sealed to away from the intercellular junction, but, are likely present within the apical membrane adjacent to the junction as well.

~9 pA at 100 mV, or 11 GΩ, events were present in MDCKI or Caco-2_{BBe} cells expressing both high and low levels of claudin-2, but were absent in cells devoid of claudin-2, i.e. the parental MDCKI line. The frequency of detection correlated with claudin-2 expression indicating that these events are strictly claudin-2 dependent. In contrast, the ~4 pA at 100 mV, or 25 GΩ, events were detected in the presence or absence of claudin-2 expression. At this point, we cannot define these events as due to a specific protein or channel. However, they are only detected at the tight junction and may, therefore represent a paracellular, i.e. tight junction, channel that accounts for the electrical conductance measured across sheets of claudin-2-deficient cells using traditional methods.

## Tight junction pore and leak pathways

In addition to the claudin-dependent size- and charge-selective pore pathway, a charge non-selective leak pathway that allows paracellular flux of small and large molecules is also present (*Anderson and Van Itallie, 2009*, *Turner, 2009*). We and others have shown that occludin- and ZO-1 are both important to leak pathway regulation (*Buschmann et al., 2013*, *Van Itallie et al., 2009*) and that, in the intestine, increased leak pathway permeability is induced by TNF via a myosin light chain kinase-dependent process (*Buschmann et al., 2013*, *Clayburgh et al., 2005*, *Van Itallie et al., 2010*, *Zolotarevsky et al., 2002*). Our in vitro studies suggest that this pathway may have an effective radius of 62.5 Å (*Buschmann et al., 2013*). We did not, however, detect a defined population of very large openings in any of our patch clamp recordings. This suggests that the leak pathway has limited or no dynamic gating, are extremely rare, or cannot be distinguished from electrode seal loss. Alternatively, some investigators have proposed that the leak pathway opens as one tight junction strand at a time, with materials trapped between strands until the next channel opens (*Sasaki et al., 2003*). In this case, it would be very difficult to detect these larger events. However, they could impact the effective conductance of claudin-2 channel opening events, and thus potentially explain the variability in amplitude that we observed. It has also been suggested that leak pathway flux occurs primarily at tricellular tight junctions. We therefore attempted to seal patch pipettes

over tricellular contacts. Unfortunately, either the geometry or small size of these regions makes it exceedingly difficult to place and seal an electrode. We therefore conclude that either a significant technical revision to our trans-tight junction patch clamp or completely different approaches will be required to perform single channel analyses of the leak pathway or tricellular tight junction.

In summary, our novel trans-tight junction patch clamp recordings reveal sub-millisecond time-scale gating of single paracellular channels that define trans-tight junction, paracellular conductance. Unlike conventional transmembrane ion channels, which regulate flux between extracellular and intracellular compartments, the channels depicted here bridge two extracellular compartments, lumen and tissue, and, therefore, represent an entirely new class of gated ion channels. Together with mutagenesis and structural analyses, the ability to detect individual conductance events, as described here, will be a critical tool in determination of molecular mechanisms that regulate channel assembly and gating. Such studies may lead to development of pharmacological means of modulating gating activity for therapeutic purposes.

# Materials and methods

### Cell culture
Madin-Darby Canine Kidney (MDCK) I cells expressing claudin-2 under control of a tet-off inducible expression system were maintained in media with 50 ng/ml of doxycycline, and claudin-2 expression was induced by culture without doxycycline for 4 days, as previously described (*Yu et al., 2009*). Human colonic Caco-2$_{BBe}$ epithelial cells, with or without stable claudin-2 knockdown, were maintained and plated as previously described (*Raleigh et al., 2011*).

### Transepithelial resistance, conductance, and potential measurements
Cells were grown on 0.33 cm$^2$ polycarbonate semi-permeable membranes with 0.4 µm pores (Corning Life Sciences, Corning, NY) and used 4 and 10 days after plating for MDCKI and Caco-2 cells, respectively. Bridges prepared using 1% agarose in Hank's balanced saline solution (HBSS; 135 mM NaCl, 0.3 mM Na$_2$HPO$_4$, 0.4 mM MgSO$_4$, 0.5 mM MgCl$_2$, 0.3 mM KH$_2$PO$_4$, 1.3 mM CaCl$_2$, 10 mM HEPES, 5 mM KOH, pH 7.4) were used. Liquid junction potentials were negligible (< 1 mV).

Bridges were connected to calomel and Ag-AgCl electrodes and a current clamp (Physiologic Instruments, San Diego, CA), as previously described, with all experiments performed at 37°C (*Weber et al., 2010*, *Yu et al., 2009*). Transepithelial resistance was determined using current clamp pulses and Ohm's law, as described (*Turner et al., 1997*). Reversal potentials (V$_{rev}$) were measured using current clamp ramps from −10 to +10 µA before and after basolateral or apical replacement of HBSS with media in which 90% or 50% of NaCl was iso-osmotically replaced by mannitol. All biionic potential measurements were performed in quadruplicate or greater in at least three independent experiments. 135 mM NaCl was substituted with 135 mM XCl, where X refers to the monovalent cations methylamine (MA$^+$), tetramethylammonium (TMA$^+$), ethylamine (EA$^+$), or *N*-methyl-D-glucamine (NMDG$^+$). Relative permeabilities (PNa$^+$/PCl$^-$) or (PX$^+$/PNa$^+$) were determined using the Goldmann-Hodgkin-Katz voltage equation, measured V$_{rev}$, and known composition of basolateral and apical solutions (*Weber et al., 2010*, *Yu et al., 2009*). Osmolarity of all buffers was confirmed using a model 3320 osmometer (Advanced Instruments, Norwood, MA). Absolute Na$^+$ permeabilities were determined from transepithelial resistance and PNa$^+$/PCl$^-$ by the Kimizuka and Koketsu method (*Kimizuka and Koketsu, 1964*, *Yu et al., 2009*) using activity coefficients of 0.755, 0.812, and 0.882 for NaCl at 135, 67.5, and 13.5 mM NaCl, respectively (*Truesdell, 1968*).

### Tight junction patch clamp
MDCKI and Caco-2$_{BBe}$ cells were used 4 and 10 days respectively after plating on shallow-walled clear polyethylene terephthalate membrane supports (0.4 µm pore size, Corning, Tewksbury MA), mounted in glass-bottom 35 mm Petri dishes. Currents were measured using an Axopatch 200B amplifier and pClamp software (Axon Instruments, Union City, CA) in voltage clamp mode with 5 kHz analog filtering. Borosilicate capillary tubes (World Precision Instrument, Sarasota, FL) were pulled to outer diameters of ~1 µm, resulting in access resistances of 2.5–3 MΩ when fire-polished. Monolayers were continuously perfused with apical and basolateral HBSS solution with 5 mM D-glucose. Gigaohm seals were obtained allowing current measurement relative to a reference electrode

without interference from apical conductances outside of the patch. Negative currents reflect cations moving in the direction towards the recording electrode.

To facilitate sealing it was helpful to stream pipette solution from the pipette as electrodes approached their points of contact. This was particularly true for Caco-2$_{BBe}$ recordings and we speculate that this prevents disordered well-developed microvilli from interfering with seal formation. Osmolarity was adjusted to 300 mOsm using mannitol to facilitate seal formation. The normal pipette solution was 135 mM NaCl, 5 mM KOH, 1 mM MgCl$_2$, 1.3 mM CaCl$_2$, 10 mM HEPES, pH 7.4, with 290 mOsm. When 4-aminopyridine, TEA, charybdotoxin, or apamin were included in the pipette, [NaCl] was reduced to maintain osmolarity. La$^{3+}$ (5 mM) and MTSET (1 mM) were added to both apical and basolateral chambers, but not included in the apical pipette solution. Pipette offsets were zeroed upon access of the recording electrode to the extracellular solution and with the electrode far from the monolayer. For local dilution potential measurements, bathing solution or pipette solution was replaced with HBSS in which 90% of the NaCl was isosmotically replaced with mannitol. For local biionic potential measurements, bathing solution was replaced isosmotically with methylamine chloride (MACl), tetramethylamine chloride (TMACl), or *N*-methyl-D-glucamine chloride (NMDGCl) as indicated. V$_{rev}$ of opening events was determined during repetitive 1 s voltage ramps from −100 to +100 mV with holding potentials of −100 mV. For the purposes of subtracting steady state conductance, ramps in which no openings were detected were subtracted from ramps which contained openings. The relatively short duration of these ramps permitted data oversampling at 100 KHz and *post hoc* data averaging down to 10 KHz. This provided a somewhat smaller amount of noise compared to the steady state recordings and allowed more accurate measurement of current reversal potentials. Patch clamp data and simulated currents were analyzed using Clampfit (Molecular Devices, Sunnyvale, CA). All points histograms were generated using 0.1 pA bin size and normalized to record duration (samples/s). Data were fit to Gaussian distributions. Secondary analyses to determine absolute event counts, open and closed durations, and event conductances were performed using Clampfit event detection software. Channel activity was expressed as open probability (NP$_o$) which was determined from the equation below (*Huang and Rane, 1993*):

$$NPo = \sum (open\ time\ x\ number\ of\ channels\ open)/(total\ time\ of\ record)$$

Liquid junction potentials (< 5 mV) were small relative to dilution potentials and not included in calculations. Single event opening and closing duration histograms were fit to single and double exponentials using the maximum likelihood method (TACFit X4.3.3 software, Bruxton Corporation, Seattle, WA). A bin size of 2 per decade was chosen to allow sufficient sampling of prolonged closed durations.

## Immunoblotting

Cell lysates were separated by SDS-PAGE and transferred to PVDF membranes, as described previously (*Weber et al., 2010*). Immunoblots were performed using antibodies to claudin-2 (Abcam, Cambridge, MA), E-cadherin (Cell Signaling Technology, Danvers, MA), or β-actin (Sigma, St. Louis, MO) followed by horseradish peroxidase-conjugated secondary antibodies (Cell Signaling Technology). Proteins were detected by enhanced chemiluminescence.

## Immunofluorescent staining and microscopy

Cultured monolayers were fixed with −20°C methanol and bis(sulfosuccinimidyl)suberate, as previously described (*Shen and Turner, 2005*). Tight junction proteins, ZO-1 and claudin-2 were immunostained using mouse anti-ZO-1 and rabbit anti-claudin-2 primary antibodies and Alexa Fluor 488 and 594 conjugated secondary antibodies (Life Technologies). Imaging was performed using an epifluorescence microscope (DM4000; Leica Microsystems, Bannockburn, IL) equipped with a 63× NA 1.32 PL APO oil immersion objective, DAPI, EGFP, and Texas Red laser aligned filter cubes (Chroma Technology, Rockingham, VT), and a Retiga EXi camera (QImaging, Surrey, BC, Canada) controlled by MetaMorph 7.5, as previously described (*Su et al., 2009*).

## Statistical analysis

Student's t-test was used to compare means. Statistical significance was designated as *p < 0.05, **p <0.01, and ***p <0.001. The Holm–Bonferroni method was used to correct for multiple comparisons. Data are shown as mean ± SEM.

## Acknowledgements

We thank Dr. Hiroyuki Sasaki, Teikyo Heisei University, for performing the freeze-fracture imaging, Ms. Amulya Lingaraju for technical assistance, and Dr. Francisco Benzanilla for insightful discussions. This work was supported by NIH grants R01DK61631, R01DK68271, K08DK088953, F32DK082134, P30DK42086, and R01DK062283.

## Additional information

### Funding

| Funder | Grant reference number | Author |
| --- | --- | --- |
| National Institutes of Health | K08DK088953 | Christopher R Weber |
| National Institutes of Health | F32DK082134 | Christopher R Weber |
| American Physiological Society | 2012 S&R Foundation Ryuji Ueno Award | Christopher R Weber |
| National Institutes of Health | R01DK062283 | Alan S L Yu |
| National Institutes of Health | R01HL125076 | Deborah J Nelson |
| National Institutes of Health | R01DK61631 | Jerrold R Turner |
| Crohn's and Colitis Foundation of America | | Jerrold R Turner |
| National Institutes of Health | R01DK68271 | Jerrold R Turner |

The funders had no role in study design, data collection and interpretation, or the decision to submit the work for publication.

### Author contributions

CRW, JRT, Conception and design, Acquisition of data, Analysis and interpretation of data, Drafting or revising the article; GHL, SD, Acquisition of data, Analysis and interpretation of data, Drafting or revising the article; YW, LS, Acquisition of data, Drafting or revising the article; ASLY, DJN, Conception and design, Analysis and interpretation of data, Drafting or revising the article

### Author ORCIDs

Christopher R Weber, http://orcid.org/0000-0002-2117-3184
Jerrold R Turner, http://orcid.org/0000-0003-0627-9455

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
