## [Decision Letter]

Thank you for submitting your work entitled "Claudin-2-dependent paracellular pores are dynamically gated" for peer review at *eLife*. Your submission has been favorably evaluated by Fiona Watt (Senior editor) and two reviewers, one of whom, Michael Fromm, has agreed to reveal his identity.

The reviewers have discussed the reviews with one another and the Reviewing editor has drafted this decision to help you prepare a revised submission.

Summary:

Most of the reviewers' criticisms (reproduced in full below) lie in the way the work is interpreted or presented. Reviewer #1 makes detailed suggestions for changes to the text, and we ask you to consider those or explain why you disagree. Several of Reviewer #2's points require clarification in the text of the manuscript.

Essential revisions:

The key points that require further experimentation are those raised by Reviewer #2. These are summarised as: validating what the capillary electrode is measuring, including freeze fracture images, and clarifying the claudin-2-independent current.

*Reviewer #1:*

This is a pioneering and exciting study which presents, for the first time, patch clamp data of a paracellular ion channel formed by a conductive tight junction protein, claudin-2. So far, it has been considered impossible to apply this technique for paracellular tight junction channels.

Since patch-clamping tight junction channels is technically innovative, the study focuses on evidence that the obtained results indeed reflect paracellular (instead of transcellular) conductances. Main experiments are performed on MDCK I cells, where claudin-2 channels are induced, but in order to assure the data another cell line with an "inverse" approach was used: Caco-2 cells, where claudin-2 is genuinely present and was knocked down.

Specific points:

1) The Abstract should provide more specific information, e.g. cell types studied, conductance and open probability of the claudin-2 channel.

2) That the channel conductance of claudin-2 in both cell systems is about 80 pS is an important result and should be presented as a main result.

3) Abstract and main text: "active gating". As far as I know there is no "active" and no "passive" gating, thus omit "active". The www tells that the expression "active gating" indeed exists but is used in a completely different field (a method to inhibit crosstalk of single avalanche diodes).

4) It is confusing that the terms "pore" and "channel" are used throughout the manuscript (including title and Abstract) as if they have the same meaning and are exchangeable. However, for the membrane channel community a pore is one part of a channel. The channel thus consists of a physical tunnel (the pore), a selectivity filter for size, a selectivity filter for charge, and a site providing time-variant permeability (if fast: gating site). A striking argument why the membranologists discriminate these two terms is the existence of two-pore channel (TPCs). Please use either expression in a specific way whenever you mean the one or the other.

5) In line with the above reasoning, I suggest to change the title which now reads "… paracellular pores are dynamically gated" to "…paracellular channels are dynamically gated", because it is not the pore which is gated, but the channel.

6) In the first paragraph of the subsection “Claudin-2 expression induces conductance events that can be detected by trans-tight junction patch clamp”: The citation of Gitter et al. 2000 does not fit here, because that paper is on much larger conductance spots caused by loss of entire cells. Instead, another global method for discriminating trans- and paracellular conductance is the two-path impedance method of the same lab.

7) In the fourth paragraph of the subsection “Claudin-2 channel behavior is consistent across different types of epithelia”: To tell that the ~8 pA claudin-2-dependent events had an average conductance of ~80 pS sounds trivial, because -100 mV was applied.

8) Figure 2: Change "Opening size" to "Current".

9) Figure 5: It would be easier to follow the stream of data if all MDCK I results are presented first and then, at the end, the Caco-2 results.

10) In the second paragraph of the subsection “Tight junction conductances are blocked by non-specific and specific inhibitors of the claudin- 2 pore”: "I66C … reduces paracellular flux of small cations (Angelow and Yu, 2009)". That paper says the opposite (Figure 3): I66C increases cation permeability.

11) In the subsection “Tight junction pore and leak pathway”: Regarding the supposed "leak pathway" it is said that no such very large openings were detected in any of the patch clamp recordings. However, the idea behind the concept of the bicellular "leak pathway" is that it opens and closes in a stochastic manner one horizonal strand after the other, so that this pathway is never completely open and thus has no "one-moment" conductance at all – solutes have to pass stepwise. Therefore, even if the patch pipet would hit a single strand leaks in an open condition, no electrical current would flow. I suggest you discuss this point.

*Reviewer #2:*

In the manuscript, the first trial of patch clamp examination at the epithelial tight junction site is described. While this study is sophisticated, it is an overstatement to assert that "pores are dynamically gated. First, the pipette used for recording is assumed to contain many claudin-2 pores. Second, the mechanical pressure of patching on the intercellular pores is quite strong at the molecular level. Third, the claudin structure does not explain the mechanism of gating. Fourth, it is recognized that in the epithelial cell sheet, channels/transporters are densely distributed along tight junctions. Claudin-2 expression may change their distribution. Fifth, the turnover and endocytosis of claudin-2 should be taken into consideration. Therefore, I recommend that you rewrite a large part of the manuscript, modestly stating the results without too much emphasis on the gating and instead discussing all possibilities.

1) The pressing question is whether the glass capillary electrode with a 1-μm diameter collects the current through many claudin-2-based channels-a process which should be described-or through other means. In this respect, the circuit should be precisely calculated.

2) The freeze fracture images are required. The authors have used an MDCK1 cell sheet with a 1500-Ωcm^2^ TER. This suggests the immaturity of the tight junction, which might contain only one or two strands at this time. The number of strands found in cells is critically important and should be further compared between MDCK1 and Caco-2 cells.

3) What mechanism induces the claudin-2-independent 4 pA current? Could you observe 4 pA current in the inhibitor assay in Figure 4? Is this 4 pA current decreased at 4 °C?

4) Why does a higher expression of claudin-2 increase only the opening probability?

5) There might be some inconsistency between the claudin expression in Figure 1 and the TER in Figure 1.

6) Could you explain the opening and closing mechanism of the claudin-2 channel using previously reported crystal structure information on claudin-15?

7) The turnover of claudin-2 should be considered.

---

## [Author Response]

Reviewer #1:

*Specific points: 1) The Abstract should provide more specific information, e.g. cell types studied, conductance and open probability of the claudin-2 channel.*

Thank you. This has been done.

*2) That the channel conductance of claudin-2 in both cell systems is about 80 pS is an important result and should be presented as a main result.*

Thank you. This has been done.

*3) Abstract and main text: "active gating". As far as I know there is no "active" and no "passive" gating, thus omit "active". The www tells that the expression "active gating" indeed exists but is used in a completely different field (a method to inhibit crosstalk of single avalanche diodes).*

Thank you. We agree and have made this change.

*4) It is confusing that the terms "pore" and "channel" are used throughout the manuscript (including title and Abstract) as if they have the same meaning and are exchangeable. However, for the membrane channel community a pore is one part of a channel. The channel thus consists of a physical tunnel (the pore), a selectivity filter for size, a selectivity filter for charge, and a site providing time-variant permeability (if fast: gating site). A striking argument why the membranologists discriminate these two terms is the existence of two-pore channel (TPCs). Please use either expression in a specific way whenever you mean the one or the other.*

Thank you. We agree and have made this change throughout.

5) In line with the above reasoning, I suggest to change the title which now reads "…paracellular pores are dynamically gated" to "…paracellular channels are dynamically gated", because it is not the pore which is gated, but the channel.

Thank you. We agree and have made this change.

*6) In the first paragraph of the subsection “Claudin-2 expression induces conductance events that can be detected by trans-tight junction patch clamp”: The citation of Gitter et al. 2000 does not fit here, because that paper is on much larger conductance spots caused by loss of entire cells. Instead, another global method for discriminating trans- and paracellular conductance is the two-path impedance method of the same lab.*

Thank you. We have now referenced Krug at al. 2009.

*7) In the fourth paragraph of the subsection “Claudin-2 channel behavior is consistent across different types of epithelia”: To tell that the ~8 pA claudin-2-dependent events had an average conductance of ~80 pS sounds trivial, because -100 mV was applied.*

We have rephrased that sentence to state: “While NP_o_ was reduced, conductance of the claudin-2-dependent events was not affected by claudin-2-knockdown (Figure 5), demonstrating that single channel conductance was not a function of claudin-2 concentration.”

*8) Figure 2: Change "Opening size" to "Current".*

Thank you. This has been corrected.

*9) Figure 5: It would be easier to follow the stream of data if all MDCK I results are presented first and then, at the end, the Caco-2 results.*

We understand your motivation in suggesting this and did initially draft the manuscript in this way. However, we view the Caco-2 data as a confirmation that claudin-2 is necessary for the currents we detected and demonstration that these events are neither cell type nor species specific. We therefore think of it as complementary to the first part of the MDCK data. However, we believe that defining the size and charge selectivity of the currents (Figure 6) and, especially, the use of the claudin-2 mutant and MTSET derivatization (Figure 7) adds a great deal of specificity and markedly solidifies our conclusion that these are claudin-2-dependent tight junction currents. These data were in the middle of the manuscript when we ordered the presentation as you suggest. Because of their importance, we really believe they should be the last data shown. We have therefore kept the Caco-2 data where they were in the initial submission.

*10) In the second paragraph of the subsection “Tight junction conductances are blocked by non-specific and specific inhibitors of the claudin- 2 pore”: "I66C … reduces paracellular flux of small cations (Angelow and Yu, 2009)". That paper says the opposite (Figure 3): I66C increases cation permeability.*

Thank you for pointing out this error. We have eliminated this sentence.

*11) In the subsection “Tight junction pore and leak pathway”: Regarding the supposed "leak pathway" it is said that no such very large openings were detected in any of the patch clamp recordings. However, the idea behind the concept of the bicellular "leak pathway" is that it opens and closes in a stochastic manner one horizonal strand after the other, so that this pathway is never completely open and thus has no "one-moment" conductance at all – solutes have to pass stepwise. Therefore, even if the patch pipet would hit a single strand leaks in an open condition, no electrical current would flow. I suggest you discuss this point.*

Thank you. We agree with most of your logic (we are not sure that anyone has defined the bicellular leak pathway, but the model you propose is reasonable) and have now discussed this in the Discussion as follows: “We did not, however, detect a defined population of very large openings […] that leak pathway flux occurs primarily at tricellular tight junctions.”

Reviewer #2:

*In the manuscript, the first trial of patch clamp examination at the epithelial tight junction site is described. While this study is sophisticated, it is an overstatement to assert that "pores are dynamically gated. First, the pipette used for recording is assumed to contain many claudin-2 pores.*

Regardless of how many pores are present in the patch, the fact that we see discreet events that begin and end tells us that these pores open and close, or, are dynamically gated. We do agree that many pores are likely present within the pipette patch area. This emphasizes the importance of the quantitative and kinetic analyses that define a specific population. We are using the definition of dynamic as an adjective, rather than a noun.

*Second, the mechanical pressure of patching on the intercellular pores is quite strong at the molecular level.*

We agree. It should, however, also be recognized that the same limitation applies to all patch clamp studies. Further, our analyses show that the charge- and size-selectivity of the channels is not affected by the patching process, i.e. these parameters are similar when measured globally without touching the cells or by patch clamp, and that the mathematical model we now include is scalable from individual events to an entire monolayer. These results indicate that we have not markedly affected channel function by patching. The one difference we did detect is the permeability of methylamine relative to Na_+_, which is greater in patch clamp measurements than in global measurements (Figure 6). We have addressed this issue in the text as: “It is, however, interesting to note that the permeability of methylamine, … charge- and size-selectivities similar to those measured across large epithelial sheets.”

*Third, the claudin structure does not explain the mechanism of gating.*

The crystal structure published has not been fully characterized and no models that explain claudin channel function have been reported. However, one could hypothesize that one or more of the unstructured loops between beta strands or the extracellular helix adjacent to strand 4 could function to close, or obstruct, the pore. Regardless of the explanation, our data demand that gating be considered by those performing structural analyses and modeling and are one more way we believe that our study advances the field.

*Fourth, it is recognized that in the epithelial cell sheet, channels/transporters are densely distributed along tight junctions. Claudin-2 expression may change their distribution.*

We are not sure about several points in the statement “it is recognized that in the epithelial cell sheet, channels/transporters are densely distributed along tight junctions.” If by“channels/transporters” the reviewer meant transmembrane ion channels (transporters), we would have to disagree, as the apical channels are predominantly located within the microvillus brush border. If reviewer intended to refer to paracellular tight junction channels, we agree that these are thought to populate the strands. The density of ‘beads’ along the strands has been measured as ~55 per linear m (Anderson. News Physiol Sci. 2001). However it is not clear if each ‘bead’ represents one, more than one, or only part of a channel. In terms of changing their distribution, we have performed FRAP analyses of claudin-2 and find that the mobile fraction is only 33% (Raleigh et al. J Cell Biol. 2011). Others have reported values below 10% (Capaldo et al. Mol Biol Cell. 2014). It is therefore unlikely that macroscopic changes in claudin-2 distribution significantly affect our results here. Molecular redistribution, such as pore disassembly, could explain the closed state with longer latency defined by our data and, interestingly, have kinetics compatible with the observed rate of FRAP recovery. We have now speculated on this in the text: “One possibility is that the prolonged state (closed_stable_) […] and we cannot exclude this as a possible regulatory mechanism.”

*Fifth, the turnover and endocytosis of claudin-2 should be taken into consideration. Therefore, I recommend that you rewrite a large part of the manuscript, modestly stating the results without too much emphasis on the gating and instead discussing all possibilities.*

The small mobile fraction of claudin-2 seen in FRAP studies effectively exclude endocytosis as a significant mechanism of channel regulation over the millisecond to second time course of the studies reported here. Further, FRAP behaviors of claudin-1 and claudin-2 are similar (Raleigh et al. J Cell Biol. 2011), and we have defined the mechanism of claudin-1 FRAP recovery to be intra-junctional diffusion (Shen et al. J Cell Biol. 2008). Finally, the half-life of claudin-2 in MDCK cells is greater than 9 hrs (van Itallie et al. J. Membrane Biol. 2004, far slower than any dynamic channel behaviors we have measured, making it difficult to speculate that either endocytosis or protein turnover are relevant to short term regulation. This is addressed in the same paragraph as the text inserted above.

*1) The pressing question is whether the glass capillary electrode with a 1-μm diameter collects the current through many claudin-2-based channels-a process which should be described-or through other means. In this respect, the circuit should be precisely calculated.*

First, to validate the capillary electrode measurements, we augmented our data set by performing more extensive analysis of the parental MDCKI line that does not express claudin-2 (Figure 5). Here, we again see that only the ~4 pA currents are detectable when claudin-2 is absent, i.e. the ~9 pA currents are not present, and that these conductance events are insensitive to traditional ion channel inhibitors as well as cooling. The primary issue here is, therefore, what is being measured by the capillary electrode when ~9 pA events are detected. As a whole, the data show that these conductances have both size- and charge-selectivity that are expected, based on measurements across larger monolayers, of claudin-2 channels. In addition, as expected for a paracellular pathway, they are symmetrical and have a reversal potential near 0. Finally, obstruction of the claudin-2 pore by a combination of site-directed mutagenesis and chemical derivatization verifies that the claudin-2 pore is essential to the genesis of these events.

A separate question that follows from the above is whether we are measuring conductance through a single channel or through several channels arranged in series. We would postulate, based both on the ultrastructural appearance of tight junction strands as well as the modeling work reported by Claude (J Membr Biol. 1978) that we are measuring channels arranged in series. If true, this would imply that the NP_o_ measurements we calculate for the pathway are far less than those for individual channels, i.e. the channels within each strand. On rare occasions we detect superimposed events, where a second ~9 pA opening event occurs when another is already present. Examples of these can be seen in Figure 4 (the amiloride, inh- 172, DIDS panel), 6B and 6E (at the right side of each trace), and 7G (several of these dual events can be seen). This indicates that at least two, and likely many more, channels are present within the 1 um pipette.

Finally, to calculate the circuit precisely, we have provided a detailed circuit diagram in a new figure (Figure 8). This figure is now described while presenting the experimental evidence for each calculation within the Discussion.

We have addressed these issues in the text as follows: “We speculate that some of the properties observed in our single channel recordings […] the entire height of the tight junction significantly underestimates the NP_o_ of individual claudin-2 channels.” Please also see the subsection entitled “Electrical analysis of trans-epithelial conductive pathways”.

*2) The freeze fracture images are required. The authors have used an MDCK1 cell sheet with a 1500-Ωcm^2^ TER. This suggests the immaturity of the tight junction, which might contain only one or two strands at this time. The number of strands found in cells is critically important and should be further compared between MDCK1 and Caco-2 cells.*

We have previously reported Caco2BBe cell monolayers to be mostly composed of 3-5 strands (Shen et al. J Cell Sci. 2006). Others have done detailed quantitative analysis of MDCK I and MDCK II cells and shown that they also have 3-5 strands and are ultrastructurally indistinguishable from one another (Stevenson et al. J Cell Biol. 1988). We have now included freeze fracture EM images of MDCK I parental cells and claudin-2-expressing MDCK I (Figure 1) as well as Caco-2BBe (Figure 5), and show that both are composed of 3-5 strands.

The new text states: “As expected based on previous comparisons of MDCKI and MDCKII cells (Stevenson, et al., 1988), which differ primarily in their expression of claudin-2, induction of claudin-2 expression did not affect tight junction ultrastructure (Figure 1).” We also added: “Nevertheless, tight junction ultrastructure is similar in MDCK and Caco-2 cells, and both are composed of three to five strands.”

*3) What mechanism induces the claudin-2-independent 4 pA current? Could you observe 4 pA current in the inhibitor assay in Figure 4? Is this 4 pA current decreased at 4 °C?*

We do not know the mechanism of the claudin-2-independent ~4 pA currents. However, they must be due to activity of something at or adjacent to the tight junction, as we do not detect these events away from the tight junction (compare Figure 4, off the tight junction, to Figure 2).

We have taken your advice and performed trans-tight junction patch clamp analyses of claudin- 2-expressing (+cldn-2) at reduced temperatures. We did try to do these analyses at 4°C, but found that the cooling apparatus circulating cold water acted as an antenna that created too much noise. Instead, we used a gravity feed that allowed us to reduce temperature to 10°-15°C while recording. We show, in Figure 4, that the number of claudin-2-dependent (~9 pA) events falls, within a single, continuous trace, as the temperature is reduced from 37°C to 15°C. The nearly two-fold NP_o_ reduction observed parallels the reduction in paracellular Na_+_ permeability that occurs after cooling to 15°C, as now shown in Figure 4 and reported previously in claudin-2-expressing MDCK cells by us (Shen et al. J Cell Biol. 2008; Figure 2B; Yu et al. J Gen Physiol. 2009; Figure 9C) as well as Martinez-Palomo et al. (J Cell Biol. 1980; Figure 12).

We also assessed the smaller, ~4 pA events using the parental MDCK I cells that completely lack claudin-2 expression. We did this because, as noted in our request for clarification, claudin-2- independent currents are best detected in the absence of the larger events, as the smaller events can be buried within the larger events. These data are shown in Figure 4.

Similar to the ~9 pA events, the ~4 pA events were unaffected by any of the inhibitor cocktails (Figure 4). In contrast to the ~9 pA events, the ~4 pA events were resistant to cold, even when chilled to 10°C (Figure 4). In this case (relative to the monolayers with claudin-2 expression), we sacrificed the ability to demonstrate a long trace during cooling for a colder steady-state temperature because the monolayers started at a lower temperature.

Thus, the simple answer to your question is no, the ~4 pA current is not decreased at 4°C. We still do not know the mechanism that generates these currents, but do know that they are claudin-2-independent and resistant to inhibition by either pharmacological agents or reduced temperature. We believe we have characterized these events to the greatest degree possible in the absence of molecular understanding of their source or tools to specifically modulate their function. We have modified the text (please see the subsection “Temperature sensitivity of claudin-2 channels is similar whether measured by trans- tight junction patch clamp or traditional global approaches”).

*4) Why does a higher expression of claudin-2 increase only the opening probability?*

We believe the reviewer is asking why single channel current amplitude is not affected by changes in claudin-2 expression. As claudin-2 channels are oligomers or polymers, reduced expression of claudin-2 would be expected to reduce the likelihood that channel complexes assemble. A reduced number of complexes would in turn reduce the frequency of channel opening events. However, if the structure and stoichiometry of the channels that are assembled is unaffected by channel number, i.e. there is not cooperativity, one would expect the result we see, i.e. that event amplitude is independent of channel number. We have modified the text to include the specific comment: “This further suggests that the number of channels, but not the open probability of individual channels, is a function of claudin-2 expression.”

*5) There might be some inconsistency between the claudin expression in Figure 1 and the TER in Figure 1.*

We believe the reviewer is asking why little claudin-2 expression is detected by western blot in the uninduced claudin-2 transfected cells (-Cldn-2) while monolayers of these cells have reduced TER relative to the parental line. This reflects known leakiness of the expression system, and is noted in the first paragraph of the Results. While the gel would be overloaded, it is possible to detect a small amount of claudin-2 expression in the –Cldn-2 monolayers.

*6) Could you explain the opening and closing mechanism of the claudin-2 channel using previously reported crystal structure information on claudin-15?*

As noted above, one could hypothesize that one or more of the unstructured loops between beta strands or the extracellular helix adjacent to strand 4 could function to close, or obstruct, the pore. Notably, our unpublished in vivo data suggest that claudin-15, while forming a Na_+_ pore like claudin-2, cannot complement claudin-2 deficiency in some in vivo contexts. We are therefore cautious in attempting to extrapolate claudin-15 structure to understand claudin-2 function.

We have now added a comment on this issue to the Discussion (“Further, as investigators in this area seek […] may reflect transient disassembly of the claudin-2 channel complex.”).

7) The turnover of claudin-2 should be considered.

The half-life of claudin-2 in MDCK cells is greater than 9 hrs (van Itallie et al. J. Membrane Biol. 2004). This is ~5 orders of magnitude slower than any dynamic channel behaviors we have measured or modeled, making it difficult to speculate that protein turnover is relevant to short term regulation. We have now noted this in the text (“However, given the absence of a significant vesicular claudin-2 pool […] we cannot exclude this as a possible regulatory mechanism.”).